# Solving Quadratic Programs via Deep Unrolled Douglas-Rachford Splitting

**Jinxin Xiong**[1,2]**, Xi Gao**[3]**, Linxin Yang**[1,2]**, Jiang Xue**[3]**, Xiaodong Luo**[1,2]**, Akang Wang**[1,2]*

[1]**School of Data Science, The Chinese University of Hong Kong, Shenzhen, China**
[2]**Shenzhen International Center for Industrial and Applied Mathematics, Shenzhen Research Institute of Big Data, China**
[3]**School of Mathematics and Statistics, Xi'an Jiaotong University, China**

*Reviewed on OpenReview: https://openreview.net/forum?id=xOfOgPnbtF*

## Abstract

Convex quadratic programs (QPs) are fundamental to numerous applications, including finance, engineering, and energy systems. Among the various methods for solving them, the Douglas-Rachford (DR) splitting algorithm is notable for its robust convergence properties. Concurrently, the emerging field of Learning-to-Optimize offers promising avenues for enhancing algorithmic performance, with algorithm unrolling receiving considerable attention due to its computational efficiency and interpretability. In this work, we propose an approach that unrolls a modified DR splitting algorithm to efficiently learn solutions for convex QPs. Specifically, we introduce a tailored DR splitting algorithm that replaces the computationally expensive linear system-solving step with a simplified gradient-based update, while retaining convergence guarantees. Consequently, we unroll the resulting DR splitting method and present a well-crafted neural network architecture to predict QP solutions. Our method achieves up to 50% reductions in iteration counts and 40% in solve time across benchmarks on both synthetic and real-world QP datasets, demonstrating its scalability and superior performance in enhancing computational efficiency across varying sizes.

## 1 Introduction

Convex Quadratic Programs (QPs) are optimization problems characterized by a convex quadratic objective function and linear constraints. These problems form a fundamental class of optimization tasks with applications spanning finance (Markowitz, 1952; Boyd et al., 2017), engineering (Garcia et al., 1989), energy systems (Frank & Rebennack, 2016), and machine learning (Cortes, 1995; Tibshirani, 1996). Additionally, convex QPs serve as the basis for constructing sequential convex approximations or relaxations of nonconvex problems, ensuring tractable subproblems while progressively approaching high-quality solutions (Boyd & Vandenberghe, 2004).

In recent years, there has been increasing interest in the development of *first-order methods* for convex QPs, with OSQP (Stellato et al., 2020) and Splitting Conic Solver (SCS) (O'Donoghue, 2021) standing out as state-of-the-art solvers. These methods are closely related to the classical Douglas-Rachford (DR) splitting algorithm (Douglas & Rachford, 1956). Notably, they require only a few matrix factorizations, which can be reused across iterations. More recently, Lu & Yang (2025) extended the Primal-Dual Hybrid Gradient (PDHG) algorithm to effectively solve convex QPs without requiring any matrix factorization. These first-order methods have demonstrated superior efficiency compared to traditional approaches like Active-Set methods (Wolfe, 1959) and Interior Point Methods (IPMs) (Nesterov & Nemirovskii, 1994), especially for large-scale problems.

---

*Corresponding Author: Akang Wang <wangakang@sribd.cn>

With the emergence of *Learn-to-Optimize* (L2O) methods, innovative strategies have been proposed to accelerate the solving of optimization problems (Chen et al., 2022; Gasse et al., 2022). By training models on diverse problem instances, L2O aims to learn problem-specific optimization strategies that surpass traditional methods in terms of efficiency and performance. In particular, the recent work of Yang et al. (2024) proposed to unroll the matrix-free PDHG algorithm in Lu & Yang (2025). This approach enables a relatively shallow network to efficiently learn the optimal QP solutions that would otherwise require thousands of PDHG iterations. However, directly unrolling DR splitting-based algorithms to efficiently learn QP solutions presents challenges, primarily due to the need to solve a linear system at each iteration. This raises the following question:

*How to unroll DR splitting-based algorithms for solving convex QPs?*

In this work, we propose unrolling a modified DR splitting algorithm in which the linear system-solving step is replaced by an equivalent least-squares problem, eliminating the need for matrix factorization. Rather than solving the least-squares problem exactly, we apply a single-step gradient descent update. Consequently, the tailored DR splitting algorithm can be seamlessly unrolled into a learnable network, delivering high-quality solutions for convex QPs. The distinct contributions of this work are as follows:

- **Unrolled DR Splitting:** We present a well-crafted neural network framework that leverages the unrolling of a modified DR splitting algorithm to efficiently solve convex QPs.

- **Convergence Guarantee:** We present the convergence properties of the modified DR splitting algorithm and demonstrate that the unrolled network can recover the optimal QP solutions.

- **Empirical Evaluation:** We validate the proposed framework through extensive experiments. The resulting solutions are employed to warm-start the QP solver SCS. The results indicate that our method significantly enhances the efficiency of solving convex QPs. In particular, our approach reduces the number of iterations by up to 50% and the solve time by up to 40%, demonstrating its effectiveness across diverse benchmarks.

## 2 Related Works

**Learning-Accelerated QP Algorithms.** Recently, machine learning techniques have been applied to develop more effective policies within algorithms, surpassing the performance of traditional hand-crafted heuristic methods. For example, Ichnowski et al. (2021); Jung et al. (2022) employed reinforcement learning to choose hyperparameters in the OSQP solver, leading to faster convergence compared to the original update rules. Venkataraman & Amos used neural networks to learn acceleration methods in fixed-point algorithms. While both approaches demonstrated improved convergence rates empirically, the convergence can not be theoretically ensured.

**Algorithm Unrolling.** Algorithm unrolling is a technique that bridges classical iterative optimization methods and deep learning by transforming the iterative steps of an optimization algorithm into the layers of a neural network. This approach enables end-to-end training, combining the strengths of both paradigms. Unrolled networks are not only interpretable, but they also tend to require fewer parameters and less training data compared to traditional deep learning models (Monga et al., 2021). By leveraging a few layers of unrolled networks, these methods can often learn solutions that would otherwise require thousands of iterations in classical algorithms. Significant success has been achieved in applications such as sparse coding (Gregor & LeCun, 2010), compressive sensing (Sun et al., 2016), inverse problems in imaging (Aljadaany et al., 2019; Liu et al., 2020; Su et al., 2024) and communication systems (Sun et al., 2023). However, these methods are not applicable to general optimization algorithms. Recently, Li et al. (2024); Yang et al. (2024) proposed unrolling the PDHG algorithm into neural networks to address general linear programs and QPs, establishing theoretical results in this context. To the best of our knowledge, such theoretical advancements have not been extended to other efficient first-order methods, such as DR splitting-based algorithms.

**Learning Warm-Starts.** An effective strategy to accelerate solution processes is to provide high-quality initializations. For example, Baker (2019); Diehl (2019); Zhang & Zhang (2022) proposed predicting initial points to warm-start the solution process for power system applications. However, these methods mainly focus on solution mappings and do not address the specific needs of warm-starting nonlinear programming algorithms. Highly related works of Sambharya et al. (2023; 2024) train fully-connected neural networks by minimizing loss after taking several steps of the fixed-point algorithms. By incorporating additional algorithmic steps following the warm-start, the predictions are refined for downstream processes. However, due to the need for vectorizing problem parameters and solving linear systems during training, this method would struggle when addressing large-scale problems. In contrast to fixed-point algorithms, Gao et al. (2024) introduced IPM-LSTM, which replaces the time-consuming process of solving linear systems in IPMs with Long Short-Term Memory (LSTM) neural networks. While this approach shows promise in providing well-centered primal-dual solution pairs for warm-starting IPM solvers, the overhead of LSTM inference limits its scalability to larger instances.

## 3 Preliminaries

### 3.1 Quadratic Programs

In this work, we consider convex QPs of the following form:

$$
\begin{aligned}
\min_{x \in \mathbb{R}^n, s \in \mathcal{K}} \quad & \tfrac{1}{2} x^\top P x + c^\top x \\
\text{s.t.} \quad & Ax + s = b, \\
& s \in \mathcal{K},
\end{aligned}
\tag{1}
$$

where $P \in \mathbb{R}^{n \times n}$, $P = P^\top \succeq 0$, $A \in \mathbb{R}^{m \times n}$, $c \in \mathbb{R}^n$, and $b \in \mathbb{R}^m$ are the problem parameters, with $x \in \mathbb{R}^n$ and $s \in \mathcal{K} := \mathbb{R}_+^m$ denoting decision variables.

### 3.2 Douglas-Rachford Splitting

For convex QPs, the following Karush-Kuhn-Tucker (KKT) conditions are necessary and sufficient for optimality:

$$
\begin{aligned}
Ax + s = b, \qquad & Px + A^\top y + c = 0, \\
s \in \mathcal{K}, y \in \mathcal{K}^*, \qquad & s \perp y,
\end{aligned}
$$

where $y$ is the dual variable and $\mathcal{K}^*$ is the dual cone to $\mathcal{K}$, defined as $\mathcal{K}^* = \{v | v^\top z \geq 0, \forall z \in \mathcal{K}\}$. Let

$$
\begin{aligned}
u &:= \begin{bmatrix} x \\ y \end{bmatrix}, \quad & M &:= \begin{bmatrix} P & A^\top \\ -A & 0 \end{bmatrix}, \\
q &:= \begin{bmatrix} c \\ b \end{bmatrix}, \quad & \mathcal{C} &:= \mathbb{R}^n \times \mathcal{K}^*.
\end{aligned}
$$

Since $P \succeq 0$, $M + M^\top \succeq 0$, i.e., $M$ is monotone. Consequently, the KKT conditions can be expressed as the following monotone inclusion problem:

$$
0 \in Mu + q + N_\mathcal{C}(u),
\tag{2}
$$

where $N_\mathcal{C}(u)$ represents the normal cone of $\mathcal{C}$, and is defined by

$$
N_\mathcal{C}(u) := \begin{cases} \{v \mid (z - u)^\top v \leq 0, \quad \forall z \in \mathcal{C}\}, & u \in \mathcal{C}, \\ \emptyset, & u \notin \mathcal{C}. \end{cases}
$$

The optimal solutions for problem (1) can thus be obtained by solving the monotone inclusion problem (2).

To address problem (2), the well-known DR splitting algorithm (Douglas & Rachford, 1956), as outlined in Algorithm 1, can be applied. In each iteration, an intermediate point $\tilde{u}^{k+1}$ is computed by solving a linear system, which is always solvable due to the full rank of $I + M$. The updated point $u^{k+1}$ is then obtained

---

**Algorithm 1** Douglas-Rachford Splitting

---

1: **Input:** $M$, $q$, $\mathcal{C}$
2: **Initialize:** $w^0 \leftarrow \mathbf{0}$
3: **for** $k = 0, 1, \cdots$ **do**
4:     $\tilde{u}^{k+1} \leftarrow (I + M)^{-1}(w^k - q)$
5:     $u^{k+1} \leftarrow \Pi_{\mathcal{C}}\left(2\tilde{u}^{k+1} - w^k\right)$
6:     $w^{k+1} \leftarrow w^k + (u^{k+1} - \tilde{u}^{k+1})$
7: **end for**
8: **Return:** $u^k := (x^k, y^k)$

---

by reflecting $\tilde{u}^{k+1}$ around $w^k$ and projecting it onto the conic set $\mathcal{C}$ using $\Pi_{\mathcal{C}}$. Finally, the iterate $w^{k+1}$ is updated by combining $w^k$, $u^{k+1}$, and $\tilde{u}^{k+1}$, balancing objective minimization with constraint satisfaction. Let $u^*$ be the optimal solution to problem (2), if it exists. Then $u^k \rightarrow u^*$ as $k \rightarrow \infty$. Additionally, the residual $\|w^{k+1} - w^k\|^2 \rightarrow 0$ at a rate of $\mathcal{O}(1/k)$ (He & Yuan, 2015; Davis & Yin, 2016).

## 4 Deep Unrolled DR Splitting

In this work, we aim to design a neural network architecture that efficiently learns high-quality solutions for convex QPs. However, directly unrolling Algorithm 1 presents challenges due to the need to solve linear systems in Step 4. To address this, we first introduce the DR-GD algorithm. Building on this foundation, we then present the unrolled neural network framework.

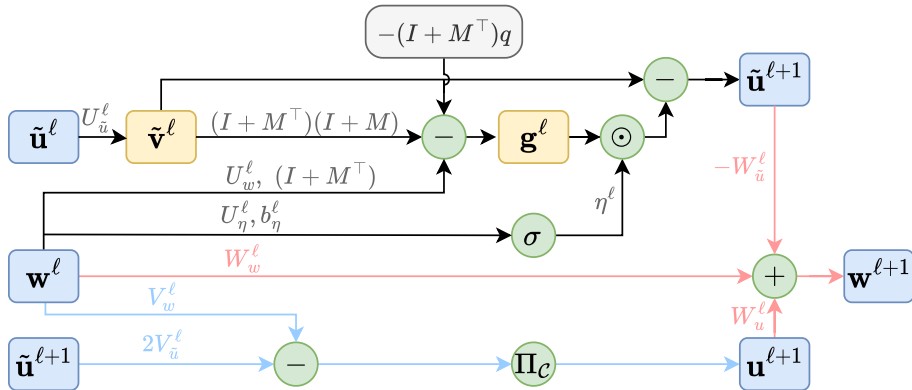

Figure 1: The figure shows one layer of the proposed DR-GD Net, which maps inputs $(\tilde{u}^\ell, w^\ell)$ to outputs $(\tilde{u}^{\ell+1}, u^{\ell+1}, w^{\ell+1})$. The update paths for $\tilde{u}, u,$ and $w$ are colored black, blue and red, respectively. Both $\tilde{u}^\ell$ and $w^\ell$ are used to compute the new state $\tilde{u}^{\ell+1}$ with parameters $\theta_{\tilde{u}}^\ell := \left(U_{\tilde{u}}^\ell, U_w^\ell, U_\eta^\ell, b_\eta^\ell\right)$. Then $\tilde{u}^{\ell+1}$ and $w^\ell$ are incorporated with ReLU activation function and parameters $\theta_u^\ell := \left(V_{\tilde{u}}^\ell, V_w^\ell\right)$ to get the updated $u^{\ell+1}$. Finally, the updated $\tilde{u}^{\ell+1}, u^{\ell+1}$ with the previous state $w^\ell$ are updated with parameters $\theta_w := \left(W_w^\ell, W_u^\ell, W_{\tilde{u}}^\ell\right)$ to produce $w^{\ell+1}$.

### 4.1 DR-GD Algorithm

Solving linear systems is often achieved via *direct methods* or *indirect methods* (Stellato et al., 2020; O'Donoghue, 2021). However, significant challenges arise when unrolling Algorithm 1 with linear systems being solved by either method.

- *Direct method:* The direct method involves matrix factorization, where dynamic pivoting is used, making the sequence of operations data-dependent and, therefore, not known in advance. This adaptive nature complicates the task of breaking the algorithm into a fixed, unrolled architecture.

---

**Algorithm 2** DR-GD Algorithm

---

1: **Input:** $M, q, \mathcal{C}$
2: **Initialize:** $w^0 \leftarrow \mathbf{0}, \tilde{u}^0 \leftarrow \mathbf{0}$
3: **for** $k = 0, 1, \cdots$ **do**
4:    $t^k \leftarrow (I + M)^\top ((I + M)\tilde{u}^k - (w^k - q))$
5:    Compute $\eta^k$ by line search
6:    $\tilde{u}^{k+1} \leftarrow \tilde{u}^k - \eta^k t^k$
7:    $u^{k+1} \leftarrow \Pi_{\mathcal{C}}\left(2\tilde{u}^{k+1} - w^k\right)$
8:    $w^{k+1} \leftarrow w^k + \left(u^{k+1} - \tilde{u}^{k+1}\right)$
9: **end for**
10: Return $u^k := (x^k, y^k)$

---

- *Indirect method:* The indirect method, in contrast, employs iterative algorithms, such as the conjugate gradient descent method, to solve the linear systems. While this approach circumvents the need for matrix factorization, it introduces the complexity of inner-outer iterations, complicating the unrolling process.

Since $I + M$ is invertible, solving the linear system in Step 4 of Algorithm 1 is equivalent to addressing the following unconstrained *least-squares problem*:

$$\min_{\tilde{u}} f(\tilde{u}) := \frac{1}{2} \left\| (I + M)\tilde{u} - (w^k - q) \right\|^2. \tag{3}$$

Moreover, provided that the $\ell^2$-norm of the error in solving the linear systems in Step 4 is finitely summable, Algorithm 1 guarantees convergence to optimal solutions (Eckstein & Bertsekas, 1992; Combettes, 2004). If problem (3) is solved to a pre-specified accuracy using iterative methods, unrolling Algorithm 1 results in a nested architecture due to the inner-outer iterations. The need to replicate the iterative process increases complexity and reduces efficiency, making the unrolled network more difficult to scale. In this work, we streamline the process by employing a single gradient step, $\tilde{u}^{k+1} \leftarrow \tilde{u}^k - \eta^k \nabla f(\tilde{u}^k)$, at each iteration, where $\tilde{u}^k$ denotes the previous iterate and $\eta^k$ is the step size. This gradient-based update enhances both efficiency and scalability when unrolled. The resulting algorithm is outlined in Algorithm 2.

**Proposition 1.** *The sequence $\{u^k\}$ generated by Algorithm 2 converges to the solution of problem (2). Furthermore, the sequences $\{x^k\}$ and $\{y^k\}$ also converge to the solution of problem (1).*

In Proposition 1, we demonstrate that Algorithm 2 converges to the optimal solution of problem (1), with the proof provided in Appendix A. While Algorithm 2 is expected to converge at a slower rate than Algorithm 1 due to the inexact evaluation involved in solving linear systems, it offers significant advantages in terms of unrolling and parameterization by eliminating the need for matrix factorizations or innerouter iterations.

## 4.2 Algorithm Unrolling

In this section, we propose DR-GD Net which is designed by unrolling Algorithm 2. The structure of the model is detailed in Algorithm 3, where parameters for channel expansion are introduced to enhance the model's flexibility and learning capacity. $\mathbf{1}^d$ is a row vector with all ones in dimension $d$, $\tilde{\mathbf{u}}^0, \mathbf{u}^0$ and $\mathbf{w}^0 \in \mathbb{R}^{(n+m) \times d^0}$. Let $\Theta := \left\{ \left\{ \theta_{\tilde{u}}^\ell, \theta_u^\ell, \theta_w^\ell \right\}_{\ell=0}^{L-1}, P_u^L \right\}$ be the parameters of the network, where $\theta_{\tilde{u}}^\ell := \left( U_{\tilde{u}}^\ell, U_w^\ell, U_\eta^\ell, b_\eta^\ell \right)$, $\theta_u^\ell := \left( V_{\tilde{u}}^\ell, V_w^\ell \right)$ and $\theta_w := \left( W_w^\ell, W_u^\ell, W_{\tilde{u}}^\ell \right)$. $\eta^\ell \in \mathbb{R}$ is the prior knowledge about the step-size of the $\ell$-th layer. $P_u^L$ is the final linear mapping for the outputs. $\sigma(\cdot)$ is the sigmoid activation function. Projection onto a product cone $\mathcal{C}$, $\Pi_{\mathcal{C}}$ is performed component-wise. For dimensions corresponding to the full real space $\mathbb{R}$, the projection is an identity mapping, leaving the values unchanged. For dimensions corresponding to the non-negative orthant $\mathbb{R}_+$, the projection is $\Pi_{\mathbb{R}_+}(a) = \max(0, a)$. This operation is precisely the definition of the ReLU activation function, making it a standard and computationally efficient building block for implementing this type of projection. The initialization can be seen as starting with $\tilde{u} = 0$, which is achieved by considering a pre-iteration state $\tilde{u}^{-1} = 0$ and $w^{-1} = q$, and running the subsequent algorithm steps for

---

**Algorithm 3** DR-GD Net

---

1: **Input:** $M, q, \mathcal{C}$, number of layers $L$ and embedding size $d^\ell$.
2: **Initialize:** $\tilde{\mathbf{u}}^0 \leftarrow \mathbf{0}, \mathbf{u}^0 \leftarrow \Pi_{\mathcal{C}}(-\mathbf{q}), \mathbf{w}^0 \leftarrow q \cdot \mathbf{1}^{d^0} + \mathbf{u}^0$;
3: **for** $\ell = 0, \cdots, L-1$ **do**
4:     Update $\tilde{u}$: $\tilde{\mathbf{v}}^\ell \leftarrow \tilde{\mathbf{u}}^\ell U_{\tilde{u}}^\ell$;
5:     $\mathbf{g}^\ell \leftarrow (I+M)^\top \left( (I+M)\tilde{\mathbf{v}}^\ell - \left( \mathbf{w}^\ell U_w^\ell - q \cdot \mathbf{1}^{d^\ell} \right) \right)$;
6:     $\tilde{\mathbf{u}}^{\ell+1} \leftarrow \tilde{\mathbf{v}}^\ell - \eta^\ell \sigma \left( \mathbf{w}^\ell U_\eta^\ell + b_\eta^\ell \right) \odot \mathbf{g}^\ell$;
7:     Update $u$: $\mathbf{u}^{\ell+1} \leftarrow \Pi_{\mathcal{C}} \left( 2\tilde{\mathbf{u}}^{\ell+1} V_{\tilde{u}}^\ell - \mathbf{w}^\ell V_w^\ell \right)$;
8:     Update $w$: $\mathbf{w}^{\ell+1} \leftarrow \mathbf{w}^\ell W_w^\ell + \left( \mathbf{u}^{\ell+1} W_u^\ell - \tilde{\mathbf{u}}^{\ell+1} W_{\tilde{u}}^\ell \right)$
9: **end for**
10: **Return** $u := \mathbf{u}^L P_u^L$

---

one iteration to generate the initial state. One layer of DR-GD Net is depicted in Figure 1. Specifically, the unrolled DR-GD Net can emulate Algorithm 2 by applying a specific instantiation of $\Theta$. Consequently, Algorithm 3 is capable of recovering the optimal solutions to problem (1) when an adequate number of layers are employed. However, by introducing learnable parameters, our goal is to allow the network to adapt to different problem instances, thereby enabling it to achieve approximate solutions with much fewer layers.

### 4.3 Training

In this work, the neural network is trained in a supervised manner, with the loss function defined as the $\ell^2$ distance between the predicted primal-dual solutions $(x, y)$ and the optimal solutions $(x^*, y^*)$, obtained by solving QPs using the SCS solver.

For a dataset of QPs, $\mathcal{M}$, we train the proposed network by finding the optimal $\Theta$ by minimizing the following loss function:

$$\min_\Theta \frac{1}{2|\mathcal{M}|} \sum_{i=1}^{|\mathcal{M}|} \left( \|x_i - x_i^*\|^2 + \|y_i - y_i^*\|^2 \right),$$

where the subscript $i$ indicates the $i^{th}$ sample in $\mathcal{M}$.

In the recent works of Sambharya et al. (2023; 2024), an unsupervised loss function based on the fixed-point residual was proposed, which eliminates the need for labeled data. However, as noted in Sambharya et al. (2024), this approach tends to focus on localized optimization metrics within the iterative process rather than the overall objective, potentially limiting its effectiveness; that is, the loss values may appear sufficiently low even when the solutions produced are far from optimal. By leveraging supervised learning with a regression loss, the DR-GD Net can learn an effective warm-start point, enhancing the performance of downstream DR splitting-based algorithms. This approach not only aligns the network architecture with the underlying algorithm but also incorporates the global insights provided by the ground truth optimal solutions.

## 5 Computational Studies

To evaluate the proposed framework, we first analyze the convergence behavior of Algorithm 2 on synthetic instances of varying sizes. Next, we compare its performance against state-of-the-art solvers and learning-based baselines on both synthetic and real-world datasets. Finally, we provide a detailed discussion of the results. The code is publicly available at `https://github.com/NetSysOpt/DR-GD.git`.

**Baseline Algorithms.** In our experiments, we denote our algorithm as `DR-GD-NN` and compare it against state-of-the-art solvers and L2O algorithms.

The solvers considered are:

(i) `SCS` (O'Donoghue, 2021): A first-order solver for quadratic cone programming based on the DR splitting algorithm with homogeneous self-dual embedding.

(ii) `OSQP` (Stellato et al., 2020): A first-order convex QP solver based on the ADMM, which is equivalent to DR splitting under appropriate variable transformations.

(iii) `rAPDHG` (Lu et al., 2024): A restarted average PDHG method (Lu & Yang, 2025) for solving convex QPs.

For `SCS`, the default linear solver is used with an absolute feasibility tolerance of $10^{-4}$, relative feasibility tolerance of $10^{-4}$, and infeasibility tolerance of $10^{-7}$. `SCS` also employs techniques such as adaptive step sizing, over-relaxation, and data normalization to enhance performance. Further details can be found in O'Donoghue (2021). `rAPDHG` is configured with absolute and relative tolerances of $10^{-4}$ and enabled $\ell_2$ norm rescaling, as it otherwise exhibited convergence difficulties.

Unlike `SCS`, `OSQP` enforces only primal and dual feasibility without explicitly verifying the termination criteria based on the primal-dual gap. To ensure a fair comparison, we follow the approach in Lu & Yang (2025); O'Donoghue (2021) to obtain `OSQP` solutions within the desired gap tolerance. Specifically, we initialize absolute and relative feasibility tolerances at $10^{-4}$ (as in `SCS`) and iteratively halve them if the gap tolerance is not met. Additionally, the convergence-checking interval is set to be 1 for both `SCS` and `OSQP` in all experiments.

The learning-based baseline algorithms include:

(i) `L2WS(Fp)` (Sambharya et al., 2024): A feedforward neural network that takes vectorized instance parameters as input to generate warm starts, followed by 60 fixed algorithmic iterations, as suggested in Sambharya et al. (2024). The network is trained with the fixed-point residual as the unsupervised loss function.

(ii) `L2WS(Reg)` (Sambharya et al., 2024): A method that shares the same architecture and number of iterations as `L2WS(Fp)` but employs a regression loss function instead.

(iii) `GNN` (Chen et al., 2024): A graph neural network trained using a regression loss on graphs representing the QP instances.

**Datasets.** The datasets used in this work include synthetic benchmarks and perturbed real-world instances. Specifically, the datasets are:

(i) **QP (RHS)** (Donti et al., 2021): Convex QPs parameterized only by the right-hand side of equality constraints, generated as in Donti et al. (2021), with $n = 200, 500, 1000$ in the experiments.

(ii) **QP** (Gao et al., 2024): The dataset was generated as in Gao et al. (2024), where all the parameters are perturbed by a random factor sampled from $U[0.9, 1.1]$.

(iii) **QPLIB** (Furini et al., 2019): Selected instances from Furini et al. (2019) with all parameters perturbed by a random factor sampled from $U[0.9, 1.1]$.

(iv) **Portfolio** (Stellato et al., 2020): Consider the portfolio optimization problem, as introduced in Stellato et al. (2020), which is formulated as follows:

$$\min_{x \in \mathbb{R}^n, y \in \mathbb{R}^k} \quad x^\top D x + y^\top y - \frac{1}{\gamma} \mu^\top x$$
$$\text{s.t.} \quad y = F^\top x$$
$$\mathbf{1}^\top x = 1$$
$$x \geq 0$$

where the variable $x \in \mathbb{R}^n$ represents the portfolio, $y \in \mathbb{R}^k$ is the axillary variable. The problem is parameterized by $\mu \in \mathbb{R}^n$ the vector of expected returns, $\gamma > 0$ the risk aversion parameter, $F \in \mathbb{R}^{n \times k}$ the factor loading matrix and $D \in \mathbb{R}^{n \times n}$ a diagonal matrix describing the asset-specific risk. The problems with $k = 100, 200, 300, 400$ factors and $n = 10k$ assets are considered in the experiments. The instances are generated by sampling $F_{ij} \sim N(0,1)$ with 50% nonzero elements, $D_{ii} \sim U[0, \sqrt{k}]$, $\mu_i \sim N(0,1)$ and $\gamma = 1$.

All instances considered in this work are formulated as in (4).

$$\begin{aligned}
\min_{x \in \mathbb{R}^n} \quad & \frac{1}{2}x^\top P x + c^\top x \\
\text{s.t.} \quad & Ax = b \\
& Gx \leq h \\
& l \leq x \leq u
\end{aligned} \tag{4}$$

where $P = P^\top \succeq 0, c \in \mathbb{R}^n, A \in \mathbb{R}^{m_1 \times n}, b \in \mathbb{R}^{m_1}, G \in \mathbb{R}^{m_2' \times n}, h \in \mathbb{R}^{m_2'}$ are the model parameters. The bounds on the variables are given by $l, u \in \mathbb{R}^n$. The formulation in problem (4) can be transformed into problem (1) by integrating the bound constraints into inequality constraints and merging the equality and inequality constraints. The size of each dataset is listed in Table 1.

Table 1: Problem sizes

| Instance | | $n$ | $m_1$ | $m_2$ |
|---|---|---|---|---|
| **QP (RHS)** | $N$ | $N$ | $N/2$ | $N/2$ |
| **QP** | $N$ | $N$ | $N/2$ | $N/2$ |
| **QPLIB** | 3913 | 300 | 61 | 600 |
| | 8845 | 1,546 | 490 | 1,848 |
| | 4270 | 1,600 | 401 | 3,202 |
| | 3547 | 1,998 | 89 | 2,959 |
| **Portfolio** | $k$ | $11k$ | $k+1$ | $20k$ |

For each dataset, 400 samples are generated for training, 40 for validation, and 100 for testing. All reported results are based on the test set.

**Evaluation Configurations.** All experiments were conducted on an NVIDIA GeForce RTX 3090 GPU and a 12th Gen Intel(R) Core(TM) i9-12900K CPU, using Python 3.9.17, PyTorch 2.0.1, SCS 3.2.6 (O'Donoghue, 2021) and OSQP 0.6.7 (Stellato et al., 2020). To demonstrate the warm-start effect across various solver configurations and ensure experimental consistency with baseline methods, we explored different parameter settings for SCS on each dataset, as detailed in Appendix B.

In all experiments, DR-GD Nets with 4 layers and embedding sizes of 128 are trained with a batch size of 2, a learning rate of $10^{-5}$, and the Adam optimizer (Kingma, 2014). The parameters $\eta^l$ are set to 0.05 for QPLIB datasets and 0.1 for the other datasets. Early stopping is employed to terminate training if the validation loss shows no improvement for 10 consecutive epochs. The model achieving the best validation performance is saved for testing.

## 5.1 Convergence of DR-GD

This section analyzes the convergence behavior of Algorithm 2, the foundation of our proposed DR-GD Net. The convergence of Algorithm 2 is expected to be slower than Algorithm 1, due to the inexact evaluation at each iteration. In Table 2, we compare the performance of Algorithm 1 and Algorithm 2 on the dataset **QP** with different sizes, with a stopping criterion of $1 \times 10^{-6}$ for the fixed-point error $\|w^{k+1} - w^k\|_2$. The

columns "Obj.", "Max Eq.", "Max Ineq.", and "Iter." report the objective value, maximum equality constraint violation, maximum inequality constraint violation, and number of iterations upon termination, respectively. The "Ratio" column shows the ratio of the number of iterations required by Algorithm 2 to those required by Algorithm 1.

The results indicate that, Algorithm 2 could produce solutions with the desired feasibility and optimality, though it generally requires more iterations than the original DR splitting algorithm. The computational overhead remains manageable with a maximum factor of 1.50 and decreases as the problem size increases. Despite its slower convergence, Algorithm 2 avoids exact computation of linear systems, making it well-suited for the development of an unrolled neural network based on this approach.

Table 2: Convergence comparison between Algorithm 1 and Algorithm 2

| Instance | | Algorithm 1 | | | | Algorithm 2 | | | | Ratio |
|---|---|---|---|---|---|---|---|---|---|---|
| | | Obj. | Max Eq. | Max Ineq. | Iter. | Obj. | Max Eq. | Max Ineq. | Iter. | |
| **QP** | 200 | $-37.126$ | $4.0 \times 10^{-10}$ | $4.6 \times 10^{-10}$ | $5,049$ | $-37.126$ | $2.4 \times 10^{-6}$ | $1.4 \times 10^{-6}$ | $7,555$ | 1.50 |
| | 500 | $-90.878$ | $1.2 \times 10^{-10}$ | $8.7 \times 10^{-11}$ | $13,653$ | $-90.878$ | $2.0 \times 10^{-6}$ | $1.6 \times 10^{-6}$ | $16,523$ | 1.21 |
| | 1000 | $-160.185$ | $3.8 \times 10^{-11}$ | $3.2 \times 10^{-11}$ | $30,844$ | $-160.185$ | $2.6 \times 10^{-7}$ | $2.0 \times 10^{-6}$ | $36,280$ | 1.18 |

## 5.2 Computational Results on DR-GD Net

**QP (RHS).** We compare the warm-start performance of `L2WS(FP)`, `L2WS(Reg)`, `GNN`, and `DR-GD-NN` on the **QP (RHS)** test set across various problem sizes. The instances are parameterized by the right-hand sides of the equality constraints, encapsulated in a single vector that serves as the input to the neural network in the `L2WS` models. The results are summarized in Table 3. The column labeled "Cold Start" shows the number of iterations and the solve time required by the SCS solver without warm starts. The "Iters."/ "Time (s)" and "Ratio" columns present the iterations/total time (including model inference and solve time in seconds) and the average reduction ratio achieved by each method across all test problems. To align the experiment settings of baseline methods, solver settings were configured with advanced techniques disabled, as shown in Table 6. The best results in each category are highlighted in bold.

Overall, `L2WS(Reg)` and `DR-GD-NN` demonstrate comparable performance. Specifically, `L2WS(Reg)` achieves the best results on instances with 200 variables, where the simple right-hand-side perturbation facilitates learning the mapping from parameters to warm-start points. However, as the problem scale increases, `DR-GD-NN` surpasses `L2WS(Reg)`, achieving a 45.4% and 53.6% reduction in iterations for datasets with 500 and 1000 variables, respectively. This highlights the efficiency of `DR-GD-NN` in handling larger instances, where the increased input size adds complexity to the solution mapping for `L2WS`. The `L2WS(FP)` variant is generally outperformed by `L2WS(Reg)`, except in the smallest case with 100 variables. This result demonstrates the superiority of using a regression loss over the unsupervised fixed-point residual loss, which primarily focuses on intermediate metrics within the iterative process. This could potentially lead `L2WS(FP)` to converge to suboptimal regions, particularly in larger and more complex problems. In contrast, the regression loss leverages global information from ground-truth solutions, enabling better performance across a broader range of problem sizes and complexities. Additionally, while `GNN` performs competitively on instances with 200 variables, it falls short on larger datasets compared to `DR-GD-NN`.

In terms of solution time, the proposed method consistently outperforms the baseline approaches, achieving reductions of up to 53.5%. Notably, even under some rare cases where the reduction in iterations is not the largest, `DR-GD-NN` still demonstrates superior efficiency. This advantage is attributed to the fast inference time of the proposed framework, which eliminates the need for the additional iterative steps required by the `L2WS` methods. For smaller instances, the reduction in the solving time is less pronounced compared to larger problems, probably because the shorter solving time for small instances leaves less room for improvement.

**QP.** In this section, we compare the performance of `GNN` and `DR-GD-NN`, both of which are trained in a supervised manner, using the loss function defined in Section 4.3. Since both methods are agnostic to the parameterization of the problems, we evaluate their performance on the dataset **QP** across different problem

Table 3: Comparison of results on **QP (RHS)** dataset between `L2WS(Fp)`, `L2WS(Reg)`, `GNN` and `DR-GD-NN`.

| Instance | | Cold Start | L2WS(Fp) | | L2WS(Reg) | | GNN | | DR-GD-NN | |
|---|---|---|---|---|---|---|---|---|---|---|
| | | Iters. | Iters. ↓ | Ratio ↑ | Iters. ↓ | Ratio ↑ | Iters. ↓ | Ratio ↑ | Iters. ↓ | Ratio ↑ |
| **QP (RHS)** | 200 | 3,855 | 3,616 | 6.2% | **2,018** | **47.6%** | 2,203 | 42.8% | 2,208 | 42.6% |
| | 500 | 10,827 | 10,820 | 0.1% | 6,117 | 43.5% | 6,350 | 41.3% | **5,907** | **45.4%** |
| | 1000 | 24,268 | 24,254 | 0.1% | 21,566 | 11.1% | 22,416 | 7.6% | **11,266** | **53.6%** |
| Instance | | Cold Start | L2WS(Fp) | | L2WS(Reg) | | GNN | | DR-GD-NN | |
| | | Time (s) | Time (s) ↓ | Ratio ↑ | Time (s) ↓ | Ratio ↑ | Time (s) ↓ | Ratio ↑ | Time (s) ↓ | Ratio ↑ |
| **QP (RHS)** | 200 | 0.470 | 0.490 | -4.5% | 0.293 | 37.7% | 0.311 | 34.6% | **0.289** | **38.1%** |
| | 500 | 6.800 | 7.348 | -8.2% | 4.436 | 34.8% | 4.019 | 40.0% | **3.735** | **44.8%** |
| | 1000 | 85.894 | 88.332 | -2.9% | 78.684 | 8.1% | 79.900 | 7.3% | **40.160** | **53.5%** |

sizes. The columns labeled "Cold Start Iters."/"Cold Start Time (s)" and "Iters."/"Time (s)" denote the average number of iterations taken by the SCS solver with cold start and warm start points provided by the two methods, respectively, using the same configuration as in the previous section. The "Ratio" column represents the average ratio of reduction in the number of iterations and total time, respectively. `GNN` and `DR-GD-NN` achieve similar performance on smaller datasets with 200 variables. However, `DR-GD-NN` performs significantly better than `GNN` on larger datasets with 500 and 1000 variables. This trend is consistent with the results in Table 3, where both methods show comparable performance on smaller problems but `DR-GD-NN` demonstrates greater scalability and efficiency in learning effective warm-start points for the SCS solver on larger-scale problems.

Table 4: Comparison of results on **QP** dataset between `GNN` and `DR-GD-NN`.

| Instance | | Cold Start | GNN | | DR-GD-NN | |
|---|---|---|---|---|---|---|
| | | Iters. | Iters. ↓ | Ratio ↑ | Iters. ↓ | Ratio ↑ |
| **QP** | 200 | 4,568 | 3,200 | 29.9% | 3,168 | **30.6%** |
| | 500 | 10,507 | 10,432 | 0.5% | 8,768 | **16.4%** |
| | 1000 | 12,660 | 12,672 | 0.3% | 10,862 | **14.1%** |
| Instance | | Cold Start | GNN | | DR-GD-NN | |
| | | Time (s) | Time (s) ↓ | Ratio ↑ | Time (s) ↓ | Ratio ↑ |
| **QP** | 200 | 0.554 | 0.427 | 23.4% | 0.406 | **26.7%** |
| | 500 | 6.591 | 6.701 | -0.19% | 5.517 | **16.1%** |
| | 1000 | 21.543 | 21.889 | -1.1% | 18.639 | **13.0%** |

**QPLIB and Portfolio.** In this section, we evaluate the performance of `DR-GD-NN` on larger datasets, including instances from QPLIB and synthetic portfolio optimization problems. For these experiments, we use the default settings of SCS, as outlined in Table 6, to highlight the practical warm-start effect of our approach.

To evaluate performance, we analyze the warm-start effect in terms of both solving time and the number of iterations. The columns labeled "OSQP", "rAPDHG" and "SCS" present the solve time and iteration count for the two solvers under cold-start conditions. The column "SCS (Warm Start)" includes the number of iterations required by SCS after utilizing the warm-start points, the inference time of `DR-GD-NN`, and the solve time, along with the total time, labeled as "Iters.", "Inf. Time (s)", "Solve Time (s)", and "Time (s)" respectively. The ratio of improvement on the total time and number of iterations are included in the column "Ratio". The observed reduction in iterations ranges from 33.7% to 57.8%, demonstrating that the warm-start points generated by `DR-GD-NN` enable robust performance, even on more challenging instances.

While the model inference time increases with problem size, it remains negligible compared to the solve time, especially for larger-scale problems. The reduction in solve time ranges from 8.5% to 55.7%. For the **Portfolio** dataset with 100 factors and the **QPLIB** instance 3913, the improvement in solve time is minimal

compared to other cases. This is attributed to the relatively small size of these problems and their inherently short solve times. Consequently, even with a significant reduction in the number of iterations, the overall improvement in solve time is limited.

Table 5: Performance of DR-GD Net on **QPLIB** and **Portfolio** datsets.

| Instance | | OSQP | rAPDHG | SCS | | SCS (Warm Start) | | | | Ratio | |
|---|---|---|---|---|---|---|---|---|---|---|---|
| | | Time (s) | Time (s) | Iters. | Time (s) | Iters. | Inf. Time (s) | Solve Time (s) | Time (s) | Iters. | Time |
| **QPLIB** | 3913 | 0.043 | 1.988 | 262 | 0.037 | 98 | 0.003 | 0.031 | **0.034** | 62.5% | 8.9% |
| | 4270 | **0.744** | 7.779 | 5,677 | 1.693 | 2,705 | 0.007 | 0.816 | 0.823 | 57.8% | 55.7% |
| | 8845 | 2.328 | 21.976 | 10,519 | 2.771 | 5,393 | 0.004 | 1.445 | **1.449** | 40.3% | 39.0% |
| | 3547 | 5.886 | 5.579 | 29,690 | 6.283 | 17,286 | 0.011 | 3.655 | **3.666** | 37.5% | 37.3% |
| **Portfolio** | 100 | 0.155 | 2.165 | 654 | 0.150 | 428 | 0.004 | 0.132 | **0.136** | 33.7% | 8.5% |
| | 200 | 0.411 | 2.372 | 997 | 0.626 | 545 | 0.005 | 0.363 | **0.368** | 45.0% | 40.8% |
| | 300 | 1.056 | 2.763 | 1,272 | 1.656 | 734 | 0.007 | 1.007 | **1.014** | 42.2% | 38.6% |
| | 400 | 2.183 | 3.481 | 1,533 | 4.280 | 712 | 0.010 | 2.140 | **2.150** | 53.4% | 49.6% |

### 5.3 Analysis of Loss and Iteration Improvement

Figure (2a) provides a detailed analysis of the relationship between validation loss and iteration improvement ratio over training epochs for the dataset **Portfolio** with 100 factors. The figure demonstrates that the validation loss steadily decreases as training progresses, indicating that the `DR-GD-NN` model is effectively learning to generate high-quality warm-start points. Simultaneously, the iteration improvement ratio increases as the loss decreases, suggesting a strong correlation between the proximity of the predicted solutions to the ground-truth solutions and the models warm-starting efficiency. This trend highlights how the predicted warm-start points, as they approach the optimal solutions, lead to a decrease in the number of SCS solver iterations required, thereby enhancing the solver's overall efficiency.

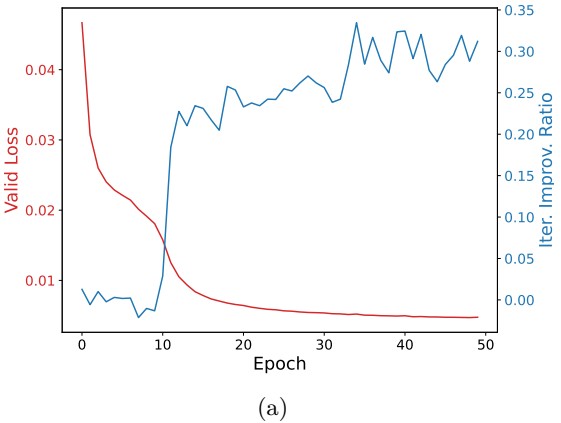
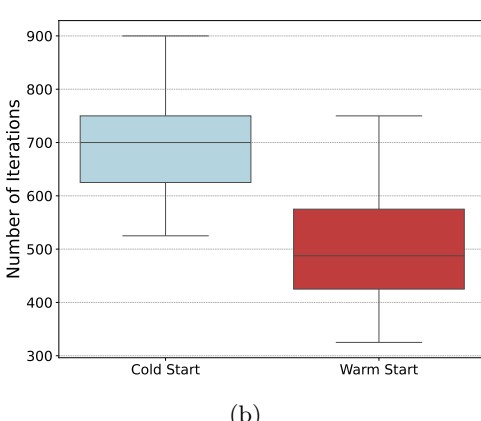

(a)                                                                 (b)

Figure 2: (a) Validation loss and iteration improvement ratio on the dataset **Portfolio** with 100 factors during training. (b) Box plot of the number of iterations required for cold starts and warm starts on the testing samples of the dataset **Portfolio** with 100 factors.

Additionally, Figure (2b) presents a box plot comparing the distribution of solver iterations required for cold starts and warm starts across all test samples. The clear separation between the distributions highlights the consistent and significant warm-start effect throughout the testing dataset. Specifically, the median number of iterations required after warm-starting is significantly lower than that for cold-starting, with a noticeably narrower interquartile range. This reduced variability suggests that `DR-GD-NN` not only enhances average solver performance but also contributes to more stable and predictable solver behavior.

# 6 Conclusions

In this work, we introduce the DR-GD Net, an unrolled neural network model based on a modified DR splitting algorithm, which eliminated the need for exact linear system solving at each iteration. The proposed architecture can effectively learn QP solutions and serves as a high-quality warm-start mechanism for SCS, a state-of-the-art first-order solver for convex QPs. Empirical results demonstrate that the proposed framework consistently reduces both the number of iterations and solve times across a range of problem instances. Future work will proceed in two main directions. First, we will focus on integrating acceleration techniques, such as the classical Anderson acceleration method, into the unrolled network to enhance its efficiency and robustness. Second, we aim to develop a more scalable training framework, potentially using unsupervised or semi-supervised learning, to handle problems with varying sizes, structures, and numerical conditions.

**Acknowledgments**

This work was supported by the National Key R&D Program of China (Grant No. 2022YFA1003900). Jinxin Xiong, Linxin Yang and Akang Wang also acknowledge support from National Natural Science Foundation of China (Grant No. 12301416), Guangdong Basic and Applied Basic Research Foundation (Grant No. 2024A1515010306), Shenzhen Science and Technology Program (Grant No. RCBS20221008093309021), and Hetao Shenzhen-Hong Kong Science and Technology Innovation Cooperation Zone Project (No. HZQSWS-KCCYB-2024016).

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

# A Proof of Proposition 1

Our goal is to prove that the sequence generated by Algorithm 2 converges to the solution of problem (2). The proof will contain two parts:

- First, we will prove that Algorithm 2 is convergent.

- Second, we will prove that Algorithm 2 converges to the solution of problem (2).

**Lemma 2.** *Let $C$ be a convex subset of a Hilbert space $H$. For any $x \in H$, a point $p \in C$ is the projection of $x$ onto $C$ (i.e., $p = \Pi_C(x)$) if and only if $p = (Id + N_C)^{-1}(x)$, where $Id$ is the identity operator on $H$ and $N_C$ is the normal cone to $C$.*

*Proof.* By definition of projection onto convex set $C$, $p = \Pi_C(x)$ if and only if $\langle x - p, y - p \rangle \leq 0, \forall y \in C$. Recall the definition of normal cone: $\forall p \in C, N_C(p) = \{z | (y - p)^\top z \leq 0, \forall y \in C\}$. Therefore,

$$\langle x - p, y - p \rangle \leq 0, \forall y \in C \Leftrightarrow (x - p) \in N_C(p).$$

Since $C$ is a convex set, the normal cone $N_C$ is a maximal monotone operator, which ensures that its resolvent $(Id + N_C)^{-1}$ is a well-defined, single-valued function.

We can now construct the chain of equivalences:

$$\begin{aligned}
p = \Pi_C(x) &\Leftrightarrow \langle x - p, y - p \rangle \leq 0, \forall y \in C \\
&\Leftrightarrow (x - p) \in N_C(p) \\
&\Leftrightarrow p \in (Id + N_C)^{-1}(x) \\
&\Leftrightarrow p = (Id + N_C)^{-1}(x)
\end{aligned}$$

$\square$

**Lemma 3.** *A single iteration in Algorithm 2, which maps $w^k$ to $w^{k+1}$, can be expressed by the operator $T_k$ as follows:*

$$w^{k+1} = T_k(w^k) = w^k + \left[ \frac{1}{2}(Id + C_{N_C}(2\Phi_k - Id))w^k - w^k \right],$$

*where*

$$\Phi_k(w) = \left( I - \eta^k (I + M)^\top (I + M) \right) \tilde{u}^k + \eta^k (I + M)^\top (w - q)$$

*and $C_{N_C}$ is the Cayley operator associated with the normal cone $N_C$.*

*Proof.* The proof proceeds by direct substitution. The Cayley operator $C_{N_C}$ is defined as $C_{N_C} = 2(Id + N_C)^{-1} - Id$. Therefore, from Lemma 1, we get $\Pi_C = \frac{1}{2}(C_{N_C} + Id)$.

Using it we can reformulate the three key steps in Algorithm 2:

$$\begin{aligned}
&\text{Step 6: } \tilde{u}^{k+1} \leftarrow \tilde{u}^k - \eta^k (I + M)^\top ((I + M)\tilde{u}^k - (w^k - q)) \\
&\text{Step 7: } u^{k+1} \leftarrow \Pi_C \left( 2\tilde{u}^{k+1} - w^k \right) \\
&\text{Step 8: } w^{k+1} \leftarrow w^k + \left( u^{k+1} - \tilde{u}^{k+1} \right)
\end{aligned}$$

By using the definition of $\Phi_k(w)$, the step 6 can be rewrite as $\tilde{u}^{k+1} = \Phi_k(w^k)$.

By rewriting the projection operator using the Cayley operator, the step 7 can be reformulated as

$$u^{k+1} = \frac{1}{2}(C_{N_C} + Id)(2\tilde{u}^{k+1} - w^k) = \frac{1}{2} \left[ (C_{N_C} + Id)(2\Phi_k - Id) \right] w^k.$$

Then, finally, step 8 can be reformulated as

$$
\begin{aligned}
w^{k+1} &= w^k + u^{k+1} - \tilde{u}^{k+1} \\
&= w^k + \frac{1}{2}\left[(C_{N_C} + Id)(2\Phi_k - Id)\right]w^k - \Phi_k(w^k) \\
&= w^k + \left[\frac{1}{2}C_{N_C}(2\Phi_k - Id)w^k + \Phi_k(w^k) - \frac{1}{2}w^k\right] - \Phi_k(w^k) \\
&= w^k + \left[\frac{1}{2}C_{N_C}(2\Phi_k - Id)w^k - \frac{1}{2}w^k\right] \\
&= w^k + \left[\frac{1}{2}(Id + C_{N_C}(2\Phi_k - Id))w^k - w^k\right] := T_k(w^k)
\end{aligned}
$$

$\square$

**Lemma 4.** *Rewrite Algorithm 2 as $w^{k+1} = T_k(w^k)$, where $T_k = \frac{1}{2}\left(Id + C_{N_C}(2\Phi_k - Id)\right)$. Assume $\cap_{k\in\mathbb{N}}\mathrm{Fix}\,T_k \neq \phi$, then $\sum_{k\in\mathbb{N}}\|T_k(w^k) - w^k\|^2 < +\infty$, which implies that $\|w^{k+1} - w^k\| \to 0$*

*Proof.* According to Lemma 3, the Algorithm 2 can be rewritten as

$$
w^{k+1} = T_k(w^k) = w^k + \left[\frac{1}{2}(Id + C_{N_C}(2\Phi_k - Id))w^k - w^k\right], \tag{5}
$$

where

$$
\Phi_k(w) = \left(I - \eta^k(I+M)^\top(I+M)\right)\tilde{u}^k + \eta^k(I+M)^\top(w-q)
$$

and $C_{N_C}$ is the Cayley operator.

Since $\|\Phi_k(w_1) - \Phi_k(w_2)\| \leq \eta^k\|I+M\|\|w_1 - w_2\|$, $\Phi_k$ is lipschitz continuous with constant $L^k = \eta^k\lambda_{\max}(I+M)$. Also, as

$$
\langle\Phi_k(w_1) - \Phi_k(w_2), w_1 - w_2\rangle = \langle\eta^k(I+M)^\top(w_1-w_2), w_1-w_2\rangle \geq \eta^k\lambda_{\min}(I+M)\|w_1 - w_2\|^2,
$$

$\Phi_k$ is strongly monotone with constant $m^k = \eta^k\lambda_{\min}(I+M)$.

$$
\begin{aligned}
&\|2\Phi_k(w_1) - w_1 - (2\Phi_k(w_2) - w_2)\|^2 \\
&= \|2(\Phi_k(w_1) - \Phi_k(w_2)) - (w_1 - w_2)\|^2 \\
&= 4\|\Phi_k(w_1) - \Phi_k(w_2)\|^2 - 4\langle\Phi_k(w_1) - \Phi_k(w_2), w_1 - w_2\rangle + \|w_1 - w_2\|^2 \\
&\leq (4(L^k)^2 - 4m^k + 1)\|w_1 - w_2\|^2
\end{aligned}
$$

If $4(L^k)^2 - 4m^k + 1 < 1$, that is $m^k > (L^k)^2$, then $2\Phi_k - Id$ is nonexpansive and contractive.

Assume that by using line-search such that $\eta^k < \frac{\lambda_{\min}(I+M)}{\lambda_{\max}^2(I+M)}$, then $2\Phi_k - Id$ is nonexpansive and thus

$$
T_k = \frac{1}{2}(Id + C_A(2\Phi_k - Id)) \in \mathcal{A}\left(\frac{1}{2}\right)
$$

is an averaged operator. As (5) taking the form of Algorithm 1.2 in Combettes (2004) with $\lambda_k = 1, e_k = 0$ and $\alpha_k = \frac{1}{2}$, according to Theorem 3.1 and Remark 3.4 in Combettes (2004) we can conclude that $\sum_{k\in\mathbb{N}}\|T_k(w^k) - w^k\|^2 < +\infty$, and thus $\|w^{k+1} - w^k\| \to 0$ $\square$

**Corollary 5** (Corollary 5.2 Combettes (2004)). *Let $\gamma \in (0,\infty)$, let $\{\nu_k\}$ be a sequence in $(0,2)$, and let $\{a_k\}$ and $\{b_k\}$ be a squence in $\mathcal{H}$. Suppose that $0 \in A + B$ is feasible, $\sum_{k\in\mathbb{N}}\nu_k(2 - \nu_k) = +\infty$, and $\sum_{k\in\mathbb{N}}\nu_k\left(\|a_k\| + \|b_k\|\right) < +\infty$. Take $x_0 \in \mathcal{H}$ and set $\forall k \in \mathbb{N}$*

$$
x_{k+1} = x_k + \nu_k R_{\gamma A}\left(2\left(R_{\gamma B}x_k + b_k\right) - x_k\right) + a_k - \left(R_{\gamma B}x_k + b_k\right)),
$$

*where $R_{\gamma A} = (Id + \gamma A)^{-1}, R_{\gamma B} = (Id + \gamma B)^{-1}$ are the resolvants for $A$ and $B$. Then $\{x_k\}$ converges weakly to some point $x \in \mathcal{H}$ and $R_{\gamma B}x \in (A + B)^{-1}(0)$.*

**Proposition 1** The sequence $\{u^k\}$ generated by Algorithm 2 converges to the solution of problem (2). Furthermore, the sequences $\{x^k\}$ and $\{y^k\}$ also converge to the solution of problem (1).

*Proof.* From (5), let $\Phi_k = R_F + \epsilon^k$, where $\epsilon^k$ represents the error induced by the inexact evaluation of the resolvent of $F$. Therefore, the Algorithm 2 can be rewritten as

$$w^{k+1} = T_k(w^k) = \frac{1}{2}(Id + C_{N_C}(2\Phi_k - Id))w^k$$

$$= \frac{1}{2}w^k + R_{N_C}(2\Phi_k - Id)w^k - \frac{1}{2}(2\Phi_k - Id)w^k$$

$$= w^k + R_{N_C}\left(2R_F(w^k) - w^k\right) - \left(R_F(w^k) + \epsilon^k\right)$$

By taking $x_k = w^k, a_k = 0, b_k = \epsilon^k, \gamma = 1, \nu_k = 1$ and let $A = R_{N_C}$ and $B = F$, Algorithm 2 is taking the form of (5). To prove the Proposition 1, we are left to show that $\sum_{k\in\mathbb{N}}\|\epsilon^k\| < +\infty$.

Let $f_k(\tilde{u}) = \frac{1}{2}\|(I + M)\tilde{u} - (w^k - q)\|^2$, line $4 - 5$ in Algorithm 2 can be written as

$$\tilde{u}^{k+1} = \tilde{u}^k - \eta^k \nabla f_k(\tilde{u}^k)$$

where the step size $\eta^k$ is chosen to satisfy the *Wolfe conditions* (Nocedal & Wright, 1999), that is

$$f_k(\tilde{u}^k) - f_k(\tilde{u}^{k+1}) \geq c_1 \eta^k \|\nabla f_k(\tilde{u}^k)\|^2 \tag{6}$$

$$\nabla f_k(\tilde{u}^{k+1})^\top \nabla f_k(\tilde{u}^k) \leq c_2 \left\|\nabla f_k(\tilde{u}^k)\right\|^2, \tag{7}$$

where $c_1$ and $c_2$ are the parameters for the line search and $0 < c_1 < c_2 < 1$.

From (7), we have

$$\left(\nabla f_k(\tilde{u}^{k+1}) - \nabla f_k(\tilde{u}^k)\right)^\top \nabla f_k(\tilde{u}^k) \leq (c_2 - 1)\left\|\nabla f_k(\tilde{u}^k)\right\|^2.$$

$f_k$ is Lipschitz continuous with constant $L = \sigma_{\max}(I + M)$, as $\|\nabla f_k(x) - \nabla f_k(y)\| \leq L\|x - y\|$. Therefore

$$-\left(\nabla f_k(\tilde{u}^{k+1}) - \nabla f_k(\tilde{u}^k)\right)^\top \nabla f_k(\tilde{u}^k)$$

$$\leq |(\nabla f_k(\tilde{u}^{k+1}) - \nabla f_k(\tilde{u}^k))^\top \nabla f_k(\tilde{u}^k)|$$

$$\leq \|(\nabla f_k(\tilde{u}^{k+1}) - \nabla f_k(\tilde{u}^k))\|\|\nabla f_k(\tilde{u}^k)\|$$

$$\leq \eta^k L\|\nabla f_k(\tilde{u}^k)\|^2,$$

which implies that

$$\eta^k \geq -\frac{(\nabla f_k(\tilde{u}^{k+1}) - \nabla f_k(\tilde{u}^k))^\top \nabla f_k(\tilde{u}^k)}{L\|\nabla f_k(\tilde{u}^k)\|^2}$$

$$\geq \frac{(1 - c_2)\left\|\nabla f_k(\tilde{u}^k)\right\|^2}{L\|\nabla f_k(\tilde{u}^k)\|^2} = \frac{1 - c_2}{L}.$$

Therefore

$$f_k(\tilde{u}^k) - f_k(\tilde{u}^{k+1}) \geq c_1 \eta^k \|\nabla f_k(\tilde{u}^k)\|^2 \geq c_1 \frac{1 - c_2}{L}\|\nabla f_k(\tilde{u}^k)\|^2.$$

Since $f_k(\tilde{u}) \geq 0$

$$1 - \frac{f_k(\tilde{u}^{k+1})}{f_k(\tilde{u}^k)} \geq c_1 \frac{1 - c_2}{L}\frac{\|\nabla f_k(\tilde{u}^k)\|^2}{f_k(\tilde{u}^k)} \geq 2c_1 \frac{1 - c_2}{L}\sigma_{\min}(I + M),$$

where $\sigma_{\min}(I + M) \geq 1$ is the minimum singular value of $(I + M)$. (As $M + M' \succeq 0$, the real parts of the eigenvalues of $M$ are nonnegative.)

The last inequality is from:

$$\nabla f_k(\tilde{u}^k) = (I + M)^\top \left((I + M)\tilde{u}^k - (w^k - q)\right)$$

$$\frac{\|\nabla f_k(\tilde{u}^k)\|^2}{f_k(\tilde{u}^k)}$$

$$= 2\frac{\left((I+M)\tilde{u}^k - (w^k - q)\right)^\top (I+M)(I+M)^\top \left((I+M)\tilde{u}^k - (w^k - q)\right)}{\|(I+M)\tilde{u}^k - (w^k - q)\|^2}$$

$$\geq 2\lambda_{\min}\left((I+M)(I+M)^\top\right), \quad \text{(rayleigh quotient)}$$

$$= 2\sigma_{\min}(I+M).$$

From Lemma 4, $\sum_{k\in\mathbb{N}} \|w^{k+1} - w^k\|^2 < +\infty$ and $\|w^{k+1} - w^k\| \to 0$. Let us further assume that $\eta^k$ also satisfies the assumption in Lemma 4, then $\exists K$, such that $\forall k > K\ \Phi_{k+1} \approx \Phi_k$ and $2\Phi_k - Id$ is contractive. Therefore, $\forall k > K, \exists 0 < c < 1$, such that $\|w^{k+1} - w^k\| \leq c\|w^k - w^{k-1}\|$ and thus $\sum_{k=K} \|w^{k+1} - w^k\| < +\infty$. As $\sum_{k\in\mathbb{N}} \|w^{k+1} - w^k\|^2 < +\infty$, $\sum_{k=0}^{K-1} \|w^{k+1} - w^k\|$ is bounded. We can conclude that $\sum_{k\in\mathbb{N}} \|w^{k+1} - w^k\| < +\infty$.

Defining $\tau^k = \frac{f_k(\tilde{u}^{k+1})}{f_k(\tilde{u}^k)}$ and taking $0 < c_1 < \frac{1}{2} < c_2 < 1$, then

$$\tau^k = \frac{f_k(\tilde{u}^{k+1})}{f_k(\tilde{u}^k)} \leq 1 - 2c_1\frac{1-c_2}{L}\sigma_{\min}(I+M) = 1 - 2c_1(1-c_2)\frac{\sigma_{\min}(I+M)}{\sigma_{\max}(I+M)} := \tau < 1.$$

Therefore

$$\|(I+M)\tilde{u}^{k+1} - (w^k - q)\| \leq \tau\|(I+M)\tilde{u}^k - (w^k - q)\|.$$

$$\|\epsilon^k\| = \|\tilde{u}^{k+1} - (I+M)^{-1}(w^k - q)\|$$

$$\leq \|(I+M)^{-1}\|\|(I+M)\tilde{u}^{k+1} - (w^k - q)\|$$

$$\leq \|(I+M)^{-1}\|\tau\|(I+M)\tilde{u}^k - (w^k - q)\|$$

$$\leq \|(I+M)^{-1}\|\tau\|(I+M)\tilde{u}^k - (w^{k-1} - q) + (w^{k-1} - q) - (w^k - q)\|$$

$$\leq \|(I+M)^{-1}\|(\tau^2\|(I+M)\tilde{u}^{k-1} - (w^{k-1} - q)\| + \tau\|w^k - w^{k-1}\|)$$

$$\leq \cdots$$

$$\leq \|(I+M)^{-1}\|(\tau^{k+1}\|(I+M)\tilde{u}^0 - (w^0 - q)\| + \sum_{j=1}^{k} \tau^{k-j+1}\|w^j - w^{j-1}\|)$$

$$:= \|(I+M)^{-1}\| \left(\sum_{j=0}^{k} \tau^{k-j+1}\|w^j - w^{j-1}\|\right), \quad \left(\text{denote } w^{(-1)} = (I+M)\tilde{u}^0\right)$$

$$:= \|(I+M)^{-1}\|a_k.$$

Next, we prove the convergence of $\sum_{k\in\mathbb{N}} \|\epsilon^k\|$ by showing the convergence of $\sum_{k\in\mathbb{N}} a_k$.

$$\sum_{k\in\mathbb{N}} a_k = \sum_{k=0}^{\infty} a_k = \sum_{k=0}^{\infty}\sum_{j=0}^{k} \tau^{k-j+1}\|w^j - w^{j-1}\|$$

$$= \sum_{j=0}^{\infty} \|w^j - w^{j-1}\|\sum_{k=j}^{\infty} \tau^{k+1-j}$$

$$= \sum_{j=0}^{\infty} \|w^j - w^{j-1}\|\tau\sum_{k=0}^{\infty} \tau^k$$

$$= \sum_{j=0}^{\infty} \|w^j - w^{j-1}\|\frac{\tau}{1-\tau} < +\infty,$$

Therefore, $\sum_{k\in\mathbb{N}} \|\epsilon^k\| \leq \sum_{k=0}^{\infty} a_k < +\infty$. Then according to Corollary 5, we can conclude that $\exists w^*$, such that $w^k \rightharpoonup w^*$. As in a finite-dimensional space strong convergence and weak convergence are equivalent, we can conclude that $w^k \to w^*$. In addition, $u^* = \tilde{u}^* = R_F(w^*) \in (F + N_{\mathcal{C}})^{-1}(0)$. $\qquad\square$

## B   SCS Settings

The parameter settings for SCS in the experiments are listed in Table 6:

Table 6: Solver configurations.

| Instance | data scaling | dual scale factor | adapt dual scale | primal scale factor | alpha | acceleration lookback | acceleration interval |
|---|---|---|---|---|---|---|---|
| **QP (RHS)/QP** | False | 1 | False | 1 | 1 | 0 | 0 |
| **QPLIB/Portfolio** | True | 0.1 | True | $10^{-6}$ | 1.5 | 10 | 10 |

## C   Offline Time

Table 7: Time for collecting training and validation instances and training model (measured in seconds).

| Instance | | Data Collection | Training |
|---|---|---|---|
| **QP(RHS)** | 200 | 6 | 215 |
| | 500 | 42 | 481 |
| | 1000 | 296 | 1007 |
| **QP** | 200 | 5 | 108 |
| | 500 | 39 | 286 |
| | 1000 | 145 | 1422 |
| **QPLIB** | 3913 | 19 | 505 |
| | 4270 | 785 | 3598 |
| | 8845 | 1355 | 5992 |
| | 3547 | 2885 | 8388 |
| **Portfolio** | 100 | 85 | 2875 |
| | 200 | 370 | 5221 |
| | 300 | 948 | 8372 |
| | 400 | 2579 | 10214 |

## D   Analysis of Number of Gradient Steps

In Table 8, we compare the number of iterations required for convergence using the same experimental settings as in Section 5.1, but on newly generated instances. The solution quality follows a similar trend to the results presented in Table 2 and is therefore omitted here for brevity. We unrolled the DR-GD algorithm (Algorithm 2) with 1, 2, 3, 4, 5, and 10 gradient steps and compared the number of iterations against the original DR splitting algorithm (Algorithm 1).

Despite the notable reduction in iterations with additional GD steps, we chose to unroll the algorithm with only a single step in the manuscript. This design was made to avoid a complicated inner-outer architecture, as mentioned in Section 4.1, and to maintain a simpler and more straightforward structure for the unrolled network.

We investigated whether unrolling more gradient descent (GD) steps per layer improves DR-GD Net performance, with results shown in Table 9. We tested versions with 1, 2, 5, and 10 inner GD steps (see Algorithm 4) on validation set of QP(RHS) and observed a clear trade-off. From the table, we observe that the most significant solution time reductions are achieved with 1 to 2 GD steps per layer and solution quality deteriorates as step count increases, evidenced by rising maximum violation values and growing $\ell_2$-error norms. These results demonstrate that our current single-step approximation yields approximate solutions of higher quality as well as better performance boost.

Table 8: Convergence comparison between Algorithm 1 and Algorithm 2 with different numbers of gradient steps per-iteration.

| Instance | | Algorithm 2 | | | | | | Algorithm 1 |
|---|---|---|---|---|---|---|---|---|
| | | step=1 | step=2 | step=3 | step=4 | step=5 | step=10 | |
| **QP** | 200 | 7447 | 6093 | 5943 | 5907 | 5876 | 5840 | 5815 |
| | 500 | 18427 | 14723 | 14171 | 14025 | 13918 | 13780 | 13646 |
| | 1000 | 22539 | 17788 | 17015 | 16817 | 16671 | 16502 | 16358 |

---

**Algorithm 4** DR-GD Net with multi-gradient steps

---

1: **Input:** $M, q, \mathcal{C}$, number of layers $L$ and embedding size $d^\ell$.
2: **Initialize:** $\tilde{\mathbf{u}}^0 \leftarrow \mathbf{0}, \mathbf{u}^0 \leftarrow \Pi_\mathcal{C}(-\mathbf{q}), \mathbf{w}^0 \leftarrow q \cdot \mathbf{1}^{d^0} + \mathbf{u}^0$, unroll_steps;
3: **for** $\ell = 0, \cdots, L-1$ **do**
4:     Update $\tilde{u}$: $w' = \mathbf{w}^\ell U_w^\ell - q \cdot \mathbf{1}^{d^\ell}$
5:     **for** $i = 1, \cdots, $ unroll_steps **do**
6:        $\tilde{\mathbf{v}}^\ell \leftarrow \tilde{\mathbf{u}}^\ell U_{\tilde{u}}^\ell$;
7:        $\mathbf{g}^\ell \leftarrow (I + M)^\top \left((I + M)\tilde{\mathbf{v}}^\ell - w'\right)$;
8:        $\tilde{\mathbf{u}}^\ell \leftarrow \tilde{\mathbf{v}}^\ell - \eta^\ell \sigma\left(\mathbf{w}^\ell U_\eta^\ell + b_\eta^\ell\right) \odot \mathbf{g}^\ell$;
9:     **end for**
10:    $\tilde{u}^{\ell+1} \leftarrow \tilde{u}^l$
11:    Update $u$: $\mathbf{u}^{\ell+1} \leftarrow \Pi_\mathcal{C}\left(2\tilde{\mathbf{u}}^{\ell+1}V_{\tilde{u}}^\ell - \mathbf{w}^\ell V_w^\ell\right)$;
12:    Update $w$: $\mathbf{w}^{\ell+1} \leftarrow \mathbf{w}^\ell W_w^\ell + \left(\mathbf{u}^{\ell+1}W_u^\ell - \tilde{\mathbf{u}}^{\ell+1}W_{\tilde{u}}^\ell\right)$
13: **end for**
14: **Return** $u \coloneqq \mathbf{u}^L P_u^L$

---

Table 9: DR-GD Net with different unrolling of gradient steps per-iteration.

| Instance | Unroll Steps | Iter. Ratio | Time Ratio | Obj. | Max Viol. | $\ell_2$ Norm |
|---|---|---|---|---|---|---|
| **QP(RHS) 200** | 1 | 40.4% | 36.0% | -36.730 | 1.264 | 0.002 |
| | 2 | 40.7% | 34.6% | -36.807 | 1.929 | 0.004 |
| | 5 | 39.7% | 29.7& | -37.120 | 1.977 | 0.004 |
| | 10 | 37.0% | 22.7% | -36.426 | 2.048 | 0.005 |
| **QP(RHS) 500** | 1 | 39.2% | 37.1% | -91.272 | 1.366 | 0.001 |
| | 2 | 39.2% | 38.2% | -91.721 | 3.477 | 0.003 |
| | 5 | 36.4% | 34.5% | -90.479 | 3.539 | 0.004 |
| | 10 | -0.8% | -4.1% | -85.441 | 29.962 | 0.479 |

# E  Testing on larger perturbations

To assess the model's robustness to larger perturbations, we evaluated a model trained with a perturbation factor of 0.1 on a test set with a 50% larger perturbation factor (0.15). As shown in Table 10, performance is maintained despite this distribution shift, providing initial evidence of the model's robustness. QPLIB 3913 was omitted from this analysis due to its limited acceleration on the original dataset.

Table 10: Testing on larger perturbations.

| Instance | | OSQP | SCS | | SCS (Warm Start) | | Ratio | |
|---|---|---|---|---|---|---|---|---|
| | | Time (s)↓ | Iters.↓ | Time (s)↓ | Iters.↓ | Time (s)↓ | Iters. ↑ | Time ↑ |
| **QPLIB** | 4270 | 0.780 | 4,577 | 1.275 | 2,270 | 0.651 | 44.8% | 42.5% |
| | 8845 | 2.391 | 10,489 | 2.292 | 6,189 | 1.767 | 32.1% | 30.9% |
| | 3547 | 4.220 | 24,201 | 5.134 | 17,073 | 3.642 | 28.9% | 28.5% |

# F  Solution Quality

The following Table 11 compares the solution quality and the inference time in seconds ("Inf. Time(s)") of baseline methods on the QP(RHS) datasets. The results demonstrate that DR-GD-NN generally achieves better objective values and smaller distances to the optimal solutions, especially for larger-sized problems. While L2WS variants do exhibit marginally better feasibility ("Max Viol."), this advantage comes at considerable computational expense ("Inf. Time") due to their mandatory feasibility restoration step.

Table 11: Solution quality of different methods on QP(RHS) datasets.

| Instance | Method | Obj. | Max Viol. | $\ell_2$ Norm | Inf. Time (s) |
|---|---|---|---|---|---|
| **QP(RHS) 200** | SCS | -36.688 | - | - | - |
| | L2WS(Reg) | -36.611 | 0.002 | 0.001 | 0.011 |
| | L2WS(Fp) | -36.437 | 0.001 | 0.007 | 0.011 |
| | GNN | -38.192 | 6.447 | 0.035 | 0.002 |
| | DR-GD-NN | -36.730 | 1.264 | 0.002 | 0.002 |
| **QP(RHS) 500** | SCS | -91.176 | - | - | - |
| | L2WS(Reg) | -90.850 | 0.001 | 0.002 | 0.072 |
| | L2WS(Fp) | -0.397 | 0.987 | 0.839 | 0.072 |
| | GNN | -66.463 | 17.903 | 0.110 | 0.005 |
| | DR-GD-NN | -91.272 | 1.366 | 0.001 | 0.004 |
| **QP(RHS) 1000** | SCS | -160.245 | - | - | - |
| | L2WS(Reg) | -84.326 | 0.008 | 0.330 | 0.303 |
| | L2WS(Fp) | -0.382 | 0.995 | 0.681 | 0.310 |
| | GNN | -70.927 | 18.181 | 0.530 | 0.018 |
| | DR-GD-NN | -160.545 | 1.750 | 0.001 | 0.012 |

# G   Supplementary Experiments on Larger Instances

In this section, we conduct experiments on QPLIB 8785, featuring problem sizes approximately 10 times larger than those in our main experiments. The specific sizes are detailed in Table 12.

The results are reported in Table 13. On average, the SCS solver required 1,559 iterations and 14.474 seconds to solve this instance. However, by using the solution from our DR-GD Net as a warm start, the iteration count and solve time for SCS were reduced to 711 and 7.025 seconds, respectively. This corresponds to an improvement of 50.7% in iterations and 49.7% in solve time. These results demonstrate that the proposed method performs effectively and maintains its benefits on significantly larger instances.

Table 12: Problem size of instance **QPLIB** 8785

| Instance | | $n$ | $m_1$ | $m_2$ |
|---|---|---|---|---|
| **QPLIB** | 8785 | 10,399 | 11,362 | 20,798 |

Table 13: Performance of DR-GD Net on **QPLIB** datsets.

| Instance | | OSQP | rAPDHG | | SCS | | SCS (Warm Start) | | | Ratio | | |
|---|---|---|---|---|---|---|---|---|---|---|---|---|
| | | Time (s) | Iters. | Time (s) | Iters. | Time (s) | Iters. | Inf. Time (s) | Solve Time (s) | Time (s) | Iters. | Time |
| **QPLIB** | 8785 | 4.255 | 15,113 | 254.6 | 1,559 | 14.474 | 711 | 0.146 | 7.025 | 7.171 | 50.7% | 49.7% |

## H    Comparison with DR-GD

The following table compares the solutions' objective values ("Obj."), feasibility satisfaction ("Max Viol.") and distances to optimal solutions ("$\ell_2$ Norm") on the QP(RHS) datasets. DR-GD($n$) represents the case where DR-GD stops after $n$ iterations, while DR-GD-NN denotes our DR-GD neural network. The results demonstrate that DR-GD-NN produces solutions with objective values ("Obj.") and proximity to optimal solutions ("$\ell_2$ Norm") quite close to those obtained from DR-GD(5000), while maintaining acceptably low constraint violations. Most notably, our DR-GD Net achieves this comparable solution quality using only 4 layers, representing a substantial improvement in computational efficiency over the conventional approach requiring 5000 iterations. These findings validate that our network architecture can effectively match the performance of standard DR-GD while requiring significantly fewer computational resources.

Table 14: Comparison of solution quality between DR-GD and DR-GD Net

| Instance | Method | Obj. | Max Viol. | $\ell_2$ Norm |
|---|---|---|---|---|
| **QP(RHS) 200** | DR-GD(4) | 6.526 | 5.101 | 0.913 |
| | DR-GD(10) | 9.615 | 3.774 | 0.995 |
| | DR-GD(100) | -18.586 | 2.178 | 0.453 |
| | DR-GD(500) | -33.726 | 0.271 | 0.052 |
| | DR-GD(1000) | -35.248 | 0.048 | 0.009 |
| | DR-GD(5000) | -35.525 | **0.000** | **0.000** |
| | DR-GD-NN | **-36.730** | 1.264 | 0.002 |
| **QP(RHS) 500** | DR-GD(4) | 19.718 | 11.194 | 1.062 |
| | DR-GD(10) | 30.545 | 8.917 | 1.167 |
| | DR-GD(100) | 1.556 | 3.449 | 0.977 |
| | DR-GD(500) | -68.233 | 1.358 | 0.287 |
| | DR-GD(1000) | -83.588 | 0.378 | 0.118 |
| | DR-GD(5000) | -92.731 | **0.009** | 0.001 |
| | DR-GD-NN | **-91.272** | 1.366 | **0.001** |
| **QP(RHS) 1000** | DR-GD(4) | 43.140 | 17.593 | 1.019 |
| | DR-GD(10) | 69.374 | 13.780 | 1.140 |
| | DR-GD(100) | 23.985 | 5.808 | 0.980 |
| | DR-GD(500) | -116.195 | 2.022 | 0.314 |
| | DR-GD(1000) | -147.401 | 0.563 | 0.143 |
| | DR-GD(5000) | -167.809 | **0.014** | 0.002 |
| | DR-GD-NN | **-160.545** | 1.750 | **0.001** |

# I   Ablation on Number of Layers

To systematically evaluate the impact of the number of layers, we trained DR-GD Net with 1 to 6 layers on the QP(RHS) datasets and measured the resulting reduction ratios in both iteration count and solve time. Figure 3 shows the resulting reduction ratios in iteration count and solve time on the validation set. The computational performance generally improves as the number of layers increases. However, the improvements become marginal beyond 4 layers. Therefore, to balance performance and model complexity, we chose to use a 4-layer architecture for our experiments.

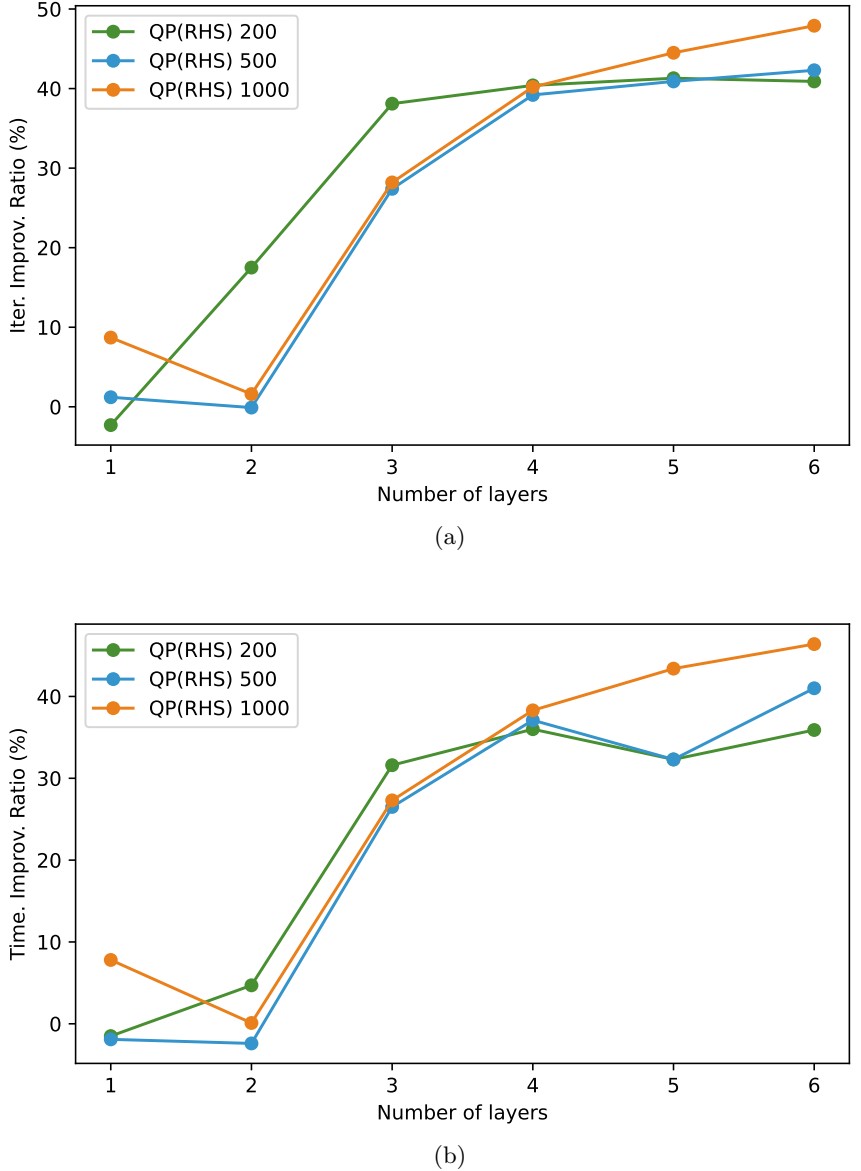

Figure 3: Improvement ratios for iteration count (3a) and solve time (3b) on the QP(RHS) validation sets, evaluated for models with different numbers of layers.

