# OpenReview forum: "Solving Quadratic Programs via Deep Unrolled Douglas-Rachford Splitting"
_TMLR — Accepted by TMLR_

### Review · Reviewer_UMYn · 2025-06-11

**Summary Of Contributions:**

The paper introduces DR‑GD Net, an unrolled neural‑network version of the Douglas–Rachford (DR) splitting method where each linear solve is replaced by a single gradient‑descent (GD) step.  This makes every DR iteration lightweight enough to serve as a network layer. The authors prove global convergence for the inexact DR‑GD algorithm, turn the unrolled iterations into a trainable architecture, and show that using the network to warm‑start the SCS solver yields sizeable speed‑ups on synthetic and real‑world quadratic‑programming (QP) benchmarks.

**Audience:**

Yes

**Broader Impact Concerns:**

No.

**Claims And Evidence:**

Yes

**Requested Changes:**

Add (at least one) stronger and more varied baselines to clarify where DR‑GD Net excels.

**Strengths And Weaknesses:**

Several strengths:

- Theory: Replacing the linear solve in DR with a single GD step is an elegant idea that enables loop‑free unrolling. Also, authors provides formal convergence proofs and demonstrates recovery of the optimal KKT point.

- Empirical Performance: Warm‑starting SCS with DR‑GD Net cuts iterations by up to 50 % and wall‑clock time by 40 %.

- Anonymous code release and complete solver hyper‑parameter listings facilitate replication.

Several weaknesses:

- Slower per‑iteration convergence of DR‑GD:  Replacing exact solves increases iterations by 18–50 % before learning. This can be viewed as a "preconditioned" PDHG? It is unclear whether DR splitting method stills has significant advantages over PDHG.

- Marginal gains on small problems: On Portfolio‑100, the solve‑time reduction is only about 8 %.

---

### Review · Reviewer_yvxR · 2025-06-13

**Summary Of Contributions:**

This paper proposes a neural network architecture based on an unrolled Douglas-Rachford (DR) splitting algorithm.  Specifically, the proposed architecture augments an inexact DR algorithm with learnable matrix-vector multiplications and nonlinearities.   The learned network (DR-GD Net) is evaluated as a warm start for SCS, a first-order solver for convex quadratic programs (QP).  Computationally, the authors demonstrate the efficacy of their approach over several existing learning-based methods.  In addition, the warm starts from the algorithm yield favorable computational results when compared to using solvers without warm starts.

**Audience:**

Yes

**Claims And Evidence:**

Yes

**Requested Changes:**

W1, W2, W4, and W5 need to be addressed.  For W3, I would appreciate it if the authors could extend their numerical results to include some of the suggested changes, in particular experiments focusing on larger instances.  Figure 1 could also be improved to be more readable and include a description.  Constantly referring back to the Figure and Algorithm 2/3 was tedious.    In addition, in the bottom left of Figure 1, I believe it should be $\tilde{\mathbf{u}}^{\ell}$ as input to match Algorithm 2/3.

**Strengths And Weaknesses:**

## Strengths
- **[S1]**: Methodologically, a strength of the paper is in the architecture design, which aligns with the iterations of the inexact DR splitting algorithm with a small number of learnable parameters.
- **[S2]**: The numerical results.  For the other learning-to-optimize-based approaches, DR-GD Net computationally finds solutions that act as a better warm start for SCS, particularly in the context of larger-sized instances.

## Weaknesses
- **[W1]**: Minimal ablation of the approach.  One of the major weaknesses is the lack of comparisons between DR-GD and DR-GD Net.  The authors claim that the motivation for DR-GD Net is that it is designed to find high-quality solutions using $L$ layers, rather than relying on the DR-GD algorithm.  However, these comparisons are never made.  Specifically, comparing the quality of the solutions for DR-GD with $L$ iterations and DR-GD Net with $L$ layers would strengthen the motivation and this claim.  Assuming DR-GD Net computes a better solution with $L$ layers, it would be helpful to assess how many iterations DR-GD takes to find a solution of similar quality.  Additionally, comparing DR-GD Net with fewer and more than $4$ layers would be a valuable ablation.  Specifically, this would provide insight into the trade-offs between network size and whether increasing the number of layers or parameters can yield better quality solutions.

- **[W2]**: No comparisons of raw solutions between methods.  While evaluating solutions produced by learn-to-optimize (L2O) methods is useful, it would be helpful to include results related to the raw solution quality of all methods compared to the solutions from solvers.

- **[W3]**: Limited benchmarks.  While the authors do compare against several baselines, there are some limitations in the benchmarks.  The first is concerning the benchmark size.  For instance, QPLIB contains much larger instances than those used for evaluation.  Given the author's claim of advantages in larger-scale instances, and QPLIB has instances that are 10 to 100 times larger, it would be helpful to evaluate on these instances.  There may even be other larger-scale QP benchmarks that would be a good testing point. In addition, exploring performance for larger deviations, e.g., beyond $U[0.9, 1.1]$, and perhaps even generalization of (reasonable) perturbations not seen during training would be informative.

 - **[W4]**: No discussion of offline time for model.  In any L2O pipeline, it is essential to have an understanding of how long data collection and training take, especially when exact solutions are used as labels.  These should be reported in the appendix at least to help readers understand the trade-offs.

- **[W5]**: Some related work on learnable variants of DR splitting/unrolling is not discussed [1,2,3].  I have not gone over these in detail, but if relevant, the authors should include a discussion on these.

## References
- [1] Raied Aljadaany, Dipan K Pal, and Marios Savvides. Douglas-rachford networks: Learning both the
image prior and data fidelity terms for blind image deconvolution. In Proceedings of the IEEE/CVF
Conference on Computer Vision and Pattern Recognition, pages 10235–10244, 2019.
- [2] Jiulong Liu, Nanguang Chen, and Hui Ji. Learnable douglas-rachford iteration and its applications in
dot imaging. Inverse Problems & Imaging, 14(4), 2020.
- [3] Yueming Su, Qiusheng Lian, Dan Zhang, and Baoshun Shi. Transformer based douglas-rachford
unrolling network for compressed sensing. Signal Processing: Image Communication, 127:117153,
2024

---

### Review · Reviewer_hB1Y · 2025-06-15

**Summary Of Contributions:**

This paper introduces a novel deep-learning framework, named DR-GD Net, for solving convex Quadratic Programs (QPs). To this end, the authors first propose a modification to the classical Douglas-Rachford (DR) splitting algorithm, called DR-GD algorithm, by replacing the computationally expensive step of solving linear system with a single gradient-descent (GD) step on regression loss. A theoretical guarantee of convergence to the optimal QP solution is proved. Then, the DR-GD algorithm is unrolled into a deep neural net architecture that can be trained on various instances of QPs in a supervised manner. The trained DR-GD Net is exploited to find a warm-starting point of a first-order QP solver (SCS). The proposed method is evaluated on several synthetic and real-world datasets and compared with other various (non-)learning-based methods.

**Audience:**

Yes

**Broader Impact Concerns:**

Since this work focuses on the theoretical aspects of an optimization algorithm and its simple applications, the Broader Impact Statement section seems unnecessary.

**Claims And Evidence:**

Yes

**Requested Changes:**

**Required Changes**

- Every point I listed above in the weaknesses should be resolved.

**Recommended Changes & Some Questions expected to be answered**

- Although it is pretty clear that the projection operator $\Pi_{\mathcal{C}}$ can be implemented with ReLU, the authors can add one more sentence to explain why.
- Have the authors tried the version of unrolling DR-GD with a few more steps of GD for each iteration of it?

**Strengths And Weaknesses:**

**Strengths**

1. The central idea of modifying a classical optimization algorithm to make it "unroll-friendly" is both novel and practical.
2. Although the theoretical guarantees for the main method (DR-GD Net) are difficult to find and verify, the method is based on a theoretically sound building block (DR-GD) having a convergence guarantee.
3. The empirical evaluation is extensive and convincing. The authors test their method on a diverse set of benchmarks.

**Weaknesses**

1. Proofs in Appendix A: statements and implications
    1. In the statement of Lemma 2, it directly equates $T_k$ to a specific form involving technical terms like $C_{N_c}$ and $\Phi_k$ *before* properly defining them in the context of Algorithm 2. While $\Phi_k$ is defined later in Equation (5), the introduction of $T_k$ seems to assume properties that are not immediately obvious from Algorithm 2. A clearer step-by-step derivation of how Algorithm 2 maps to this form of $T_k$ would improve clarity.
    2. As far as I understood, the proofs in Appendix A do not provide a non-asymptotic convergence rate.
2. Possible difficulty in scaling to larger problems
    1. A major limitation of the proposed method is that it requires the ground-truth optimal solutions of training instances. To this end, you still need to run a linear system solving algorithm or many iterations of a gradient-based algorithm that approximates it. Although the DR-GD eliminates the steps of solving linear systems inside the algorithm, the supervised training of DR-GD still suffers from them, which is nearly prohibitive for high-dimensional problems.
    2. Another problem is that a single DR-GD Net can solve issues of fixed dimension. If the problem dimension increases, you have to train a totally new DR-GD Net from scratch.
3. Ambiguity in the description of Algorithm 3 (DR-GD Net)
    1. The algorithm specifies initializing iterates ${\bf u}^0$ and ${\bf w}^0$ using ${\bf q}$ and $q$. There is no rationale provided for why these specific initializations (${\bf u}^0 \leftarrow \Pi_{\mathcal{C}}(-{\bf q})$ and ${\bf w}^0 \leftarrow q\cdot{\bf}1^{d^0}+{\bf u}^0$) were chosen or whether they are critical to the model's performance.

---

### Decision · Action_Editor_39vJ · 2025-07-20

**Recommendation:** Accept as is

**Audience:**

Yes

**Audience Explanation:**

Convex quadratic programming is widely applicable.

**Claims And Evidence:**

Yes

**Claims Explanation:**

The performance of the proposed algorithm is provided with theoretical guarantee and experimental evidence.